# Evaluation of underweight status may improve identification of the highest-risk patients during outpatient evaluation for pulmonary tuberculosis

**Peter J. Kitonsa**[1]*, **Annet Nalutaaya**[1], **James Mukiibi**[1], **Olga Nakasolya**[1], **David Isooba**[1], **Caleb Kamoga**[1], **Yeonsoo Baik**[2], **Katherine Robsky**[2], **David W. Dowdy**[1,2], **Achilles Katamba**[1,3], **Emily A. Kendall**[1,4]

1 Uganda Tuberculosis Implementation Research Consortium, Makerere University, College of Health Sciences, Kampala, Uganda, 2 Department of Epidemiology, Johns Hopkins Bloomberg School of Public Health, Baltimore, Maryland, United States America, 3 Department of Medicine, College of Health Sciences, Makerere University, Upper Mulago Hill, Kampala, Uganda, 4 Division of Infectious Diseases and Center for Tuberculosis Research, Department of Medicine, Johns Hopkins University, Baltimore, Maryland, United States of America

* kitonsap@gmail.com

**Data Availability Statement:** The data underlying the results presented in the study are available from this link: https://doi.org/10.7281/T1/7WS8AD.

## Abstract

### Background

When evaluating symptomatic patients for tuberculosis (TB) without access to same-day diagnostic test results, clinicians often make empiric decisions about starting treatment. The number of TB symptoms and/or underweight status could help identify patients at highest risk for a positive result. We sought to evaluate the usefulness of BMI assessment and a count of characteristic TB symptoms for identifying patients at highest risk for TB.

### Methods

We enrolled adult patients receiving pulmonary TB diagnoses and a representative sample with negative TB evaluations at four outpatient health facilities in Kampala, Uganda. We asked patients about symptoms of chronic cough, night sweats, chest pain, fever, hemoptysis, or weight loss; measured height and weight; and collected sputum for mycobacterial culture. We evaluated the diagnostic accuracy (for culture-positive TB) of two simple scoring systems: (a) number of TB symptoms, and (b) number of TB symptoms plus one or more additional points for underweight status (body mass index [BMI] $\leq$ 18.5 kg/m$^2$).

### Results

We included 121 patients with culture-positive TB and 370 patients with negative culture results (44 of whom had been recommended for TB treatment by evaluating clinicians). Of the six symptoms assessed, the median number of symptoms that patients reported was two (interquartile range [IQR]: 1, 3). The median BMI was 20.9 kg/m$^2$ (IQR: 18.6, 24.0), and 118 (24%) patients were underweight. Counting the number of symptoms provided an area

**Funding:** US National Institute of Health (R01HL138728 to DWD and K08AI127908 to EAK) https://grants.nih.gov/grants/oer.htm NO - The funders had no role in the study design, data collection and analysis, decision to publish, or preparation of the manuscript.

**Competing interests:** NO: The authors have declared that no competing interests exist.

under the Receiver Operating Characteristic curve (c-statistic) of 0.77 (95% confidence interval, CI: 0.72, 0.81) for identifying culture-positive TB; adding two points for underweight status increased the c-statistic to 0.81 (95%CI: 0.76, 0.85). A cutoff of $\geq 3$ symptoms had sensitivity and specificity of 65% and 74%, whereas a score of $\geq 4$ on the combined score ($\geq 2$ symptoms if underweight, $\geq 4$ symptoms if not underweight) gave higher sensitivity and specificity of 69% and 81% respectively. A sensitivity analysis defining TB by Xpert MTB/RIF status produced similar results.

## Conclusion

A count of patients' TB symptoms may be useful in clinical decision-making about TB diagnosis. Consideration of underweight status adds additional diagnostic value.

## Introduction

Each year, approximately 10 million people develop tuberculosis (TB), of whom 1.5 million die; low and middle income countries of Asia and Africa bear the largest burden of disease [1]. In ongoing efforts to end the global TB epidemic, a core strategic pillar is patient-centered care including early diagnosis, treatment, and patient support for all people with TB [2].

Several challenges limit the prompt diagnosis and treatment of TB in high-burden settings. In some instances, people with TB [3] or their healthcare providers [4] may not pursue TB testing. Even when testing occurs, it may not provide a prompt diagnosis for reasons that include equipment downtime [5], long turnaround times for off-site testing [6], or the suboptimal sensitivity of even the most sensitive rapid diagnostic tests [7]. Delays in obtaining a diagnostic result contribute to high risks of pretreatment loss to follow-up [8]. For patients or settings in which the risk of loss to follow-up is high and rapid bacteriological diagnosis is impractical, clinicians must often decide whether to start treatment empirically for the highest risk patients based on clinical presentation alone [9].

Most discussions of clinical decision-making for TB center on risk factors for developing TB and on the presence and severity of characteristic symptoms. Body mass index (BMI) is often not given the same consideration but may be useful in identifying high-risk patients. Underweight status (BMI $\leq 18.5$ mg/kg$^2$), is known to be associated with TB [10–12]: TB itself can cause weight loss, and multiple conditions associated with low body weight (malnutrition, advanced HIV) are also risk factors for TB [13], but it is not clear to what extent BMI assessment provides additional information independent of weight loss or other constitutional symptoms. To date, clinical consideration of underweight status has been limited; for example, it has been proposed as part of a multi-component prediction score to prioritize patients for TB testing in an HIV clinic population in South Africa [14], but BMI is not routinely calculated in most resource-limited settings. Determination of BMI is inexpensive and could, if shown to be sufficiently valuable, be incorporated into routine triage. In addition, while it might be expected that those with a greater number of characteristic TB symptoms are more likely to have TB, data to support this relationship are limited [15], and the degree of independence between TB symptoms and BMI is uncertain. Therefore, within a general clinical population undergoing evaluation for possible pulmonary TB, we sought to evaluate the usefulness of BMI and a count of characteristic TB symptoms, alone or in combination, for identifying patients at highest risk for TB.

## Materials and methods

### Study setting and design

We conducted a case-control study comparing patients with and without culture-positive TB, among patients undergoing TB diagnostic evaluation at four TB Diagnostic and Treatment Units (DTUs) in Kampala, Uganda. These facilities included one large public clinic, two smaller private clinics, and an HIV clinic, and collectively their patient population is fairly representative of outpatient TB diagnosis in urban Uganda. TB evaluation at these facilities typically included sputum Xpert MTB/RIF testing [Cepheid, Inc] according to routine clinical laboratory procedures, with some patients diagnosed based on sputum smear microscopy or clinical judgment.

### Study participants

The study population consisted of patients aged ≥15 years who underwent diagnostic evaluations for possible pulmonary TB at the DTUs (and who also met residence-based eligibility criteria for an ongoing study of local TB transmission, which limited enrollment at most participating health facilities to residents of certain nearby zones). After a two-month pilot period to refine study questionnaires and procedures, we recruited all eligible patients who received a TB diagnosis (regardless of whether the diagnosis was bacteriological or clinical) between 22[nd] May 2018 and 29[th] February 2020. For each enrolled patient with a TB diagnosis, we also enrolled two individuals with negative TB diagnostic evaluations (including at least one negative sputum bacteriologic result, and a decision by the clinician not to recommend treatment) at the same facility, randomly selected from among the eligible TB-negative individuals who completed their TB evaluations on an arbitrarily-selected day after the TB patient was enrolled; if the selected TB-negative participant did not return for their result and could not be contacted, then another TB-negative individual was selected using the same procedure. Participants were enrolled on the day that a decision to treat or not treat for TB was made, or as soon as possible thereafter. For the current analysis, we classified participants' "true" TB status based on the outcome of a sputum mycobacterial culture that we performed at the time of enrollment (S1 Fig).

### Data collection process

All participants completed an interview, height and weight measurement, and sputum culture. The interview included sociodemographic information, potential TB risk factors, and a standard orally-administered questionnaire (S1 Instrument) about the presence and duration of each of six TB symptoms: cough (classified as "chronic" if it had lasted at least two weeks), fevers, unexplained weight loss, chest pain, hemoptysis, and drenching night sweats. Research staff measured all participants' weight (in kg) and height (in meters) using a SECA-216 weighing scale and stadiometer (Seca Industries, Hamburg). An additional expectorated sputum specimen collected on the day of enrollment was cultured for mycobacteria using standard solid (Lowenstein Jensen (LJ) media) and liquid (Mycobacteria Growth Indicator Tube (MGIT, Beckton Dickinson)) culture methods [16]. Positive culture growth was confirmed as *Mycobacterium tuberculosis* (MTB) using MTP64 antigen testing (SD BIOLINE, Abbott Laboratories, Chicago), and patients with an MTB-positive culture who had not been diagnosed at initial TB evaluation were notified and referred to treatment. All patients who were evaluated for TB also underwent routine HIV testing (Abbott Determine [Abbott Laboratories, Chicago]) per national Ministry of Health guidelines.

## Study variables

Our outcome variable was bacteriologic TB as determined by sputum mycobacterial culture. Patients were considered culture-positive if they had MTB growth on either LJ or MGIT. Patients with two negative cultures, one negative and one contaminated culture, or only non-tuberculous mycobacterial growth were considered culture-negative, and those with both contaminated LJ and contaminated MGIT were excluded from the primary analysis (S1 Fig). In a sensitivity analysis, we used Xpert rather than culture to classify true TB status, and we excluded participants with no Xpert result.

The predictor variables considered were body mass index (BMI), and either individual TB symptoms or the total number of reported symptoms. BMI was computed as weight (kg) divided by height (meters squared) and categorized as underweight (less than 18.5 kg/m$^2$), normal weight (18.5 to 24.9 kg/m$^2$) or above normal (over 25 kg/m$^2$): in which we combined both the overweight and obese individuals due to the low individual sample sizes [15, 17, 18]. The number of reported symptoms was tallied from among chronic cough, fevers, night sweats, chest pain, weight loss and hemoptysis, for a possible count ranging from zero to six symptoms. Co-variates of age, sex, education level, employment status, cigarette smoking, alcohol consumption, and HIV status were selected a priori for inclusion in multivariable models. The primary analysis excluded participants with uninterpretable culture results (as described above) or clearly erroneous or missing height or weight measurements (n = 3; S1 Fig); no symptom or covariate data were missing.

## Statistical analysis

Categorical variables were compared using Fisher's exact test for 2x2 comparisons. Using logistic regression, we first evaluated the univariable associations of culture-positive TB with the predictors and covariates of interest, as listed above. We then developed a multivariable logistic regression model to explore the relationship between BMI, individual TB symptoms, and TB status. Starting with the covariates specified above, we performed logistic regression with backward selection at a stay significance level of >0.2. We also verified that forward selection resulted in selection of the same set of covariates for the final model.

We then evaluated the diagnostic accuracy of symptom-based, BMI-based, and combined approaches to identifying the individuals most likely to have TB among patients also selected for TB diagnostic evaluation. In particular, we estimated an area under the Receiver Operator Characteristic curve (c-statistic) for predicting TB within our enrolled population, for each of three diagnostic scoring systems: (a) Underweight BMI as a binary classifier, (b) a score from zero to six corresponding to the total number of symptoms that a patient reported, or (c) a score consisting of the sum of a patient's number of symptoms and one or more additional points for being underweight. We also estimated the sensitivity and specificity (each with binomial confidence intervals) for individual cutoffs of each score.

Our sample size was intended to provide 80% power to detect a 10% difference in c-statistic between diagnosis of TB based on number of symptoms alone and based on the combination of number of symptoms with underweight BMI with a 95% confidence [19]. All analysis was performed in Stata version 13.0, using the 'diagt', 'roctab' and 'roccomp' packages for analyses of diagnostic accuracy [20].

Because our study design enrolled only two patients with negative TB evaluations per patient diagnosed with TB (when in fact more than two patients tested negative for each patient diagnosed with TB), we used review of the diagnostic register to estimate the TB prevalence in the underlying patient population from which our participants were drawn, so that we could estimate positive predictive values (PPVs) and negative predictive values (NPVs) in that

population. First, we calculated the prevalence of sputum culture positivity among the patients whom we enrolled with a TB diagnosis, and among those whom we enrolled as TB-negative controls. We then reviewed the presumptive TB registers at participating health facilities to determine the number of negative TB evaluations per diagnosed TB patient. Applying this number as a weight to our study participants with negative TB evaluations, we calculated a weighted average that estimated the prevalence of culture-positive TB among all presumptive TB patients eligible for our study. We also estimate PPVs and NPVs in hypothetical patient populations with higher or lower TB prevalence.

### Ethical considerations

The study was approved by the Higher Degrees, Research and Ethics Committee of the Makerere University School of Public Health, Kampala-Uganda (Study Protocol Number 544). Participants provided informed consent (or assent and parental consent for those 15–17 years old) for all study activities.

## Results

### Participant characteristics and culture-positive TB

BMI, TB symptoms, and culture status were determined for 121 patients with culture-positive TB and 370 patients with negative culture results. The median age of the study participants was 32 years (IQR: 25, 41). Of the 491 participants, 50% (243/491) were female, 62% (302/491) had at least a PLE certificate (primary education), 26% (125/491) were formally employed, 26% (129/491) actively smoked cigarettes, and 13% (64/491) consumed at least one alcoholic drink per week (Table 1). Of the culture-positive patients, 89% (108/121) had a positive Xpert result, 3% (3/121) had no Xpert result but positive sputum smear microscopy, 7% (9/121) were diagnosed clinically (6 of whom had no Xpert test done due to equipment malfunction, reagent stock out or failure to expectorate), and 5% (6/121) had a negative Xpert and were recommended for treatment only after receiving the positive culture result. Of the culture negative patients, 8% (28/370) had a positive Xpert result, 7% (24/370) were recommended for TB treatment on the basis of clinical judgment (14 of whom had no Xpert test done), and 88% (326/370) were negative on Xpert and not recommended for treatment.

As determined by culture results, the prevalence of pulmonary TB among the study participants was 25% (95% confidence interval [95%CI]: 21, 29). We estimated that the TB prevalence in the underlying patient population (i.e., including individuals who presented for evaluation but were not diagnosed with TB and not selected as controls) was 12%.

Culture-positive participants were similar in age to culture negative participants (median age 32, IQR: 25, 41) but were more likely to be male (odds ratio: 2.2, 95%CI: 1.5, 3.4). Nearly half of patients (239/491) reported weight loss, only 38% (90/239) of whom had BMI ≤18.5 kg/m$^2$. Culture-positive participants had a lower median BMI (18.4kg/m$^2$, IQR: 16.9, 20.7) than culture-negative participants (21.8kg/m$^2$, IQR: 19.6, 24.3): culture-positive participants were also more likely to present with the symptom weight loss, night sweats or chronic cough at evaluation. Compared to participants of normal weight, underweight participants had 4.8 times (95%CI: 3.0, 7.7) higher odds of culture-positive TB; this association persisted (adjusted odds ratio 3.0, 95%CI: 1.8, 5.1) after adjusting for cigarette smoking, and presenting TB symptoms of weight loss, chronic cough, fevers and night sweats (Table 1). In addition to underweight BMI, each of chronic cough, weight loss, and night sweats was also significantly associated with TB in the final model, with estimated odds ratios >2 and p < 0.05.

**Symptom count, body mass index and culture-positive TB.** The most common symptoms—chronic cough and weight loss—were present in 88% and 78% of culture-positive

**Table 1. Association of clinical and sociodemographic characteristics with TB among presumptive TB patients presenting to four clinics in Kampala, Uganda.** (n = 491).

| Variable | Culture Status n (%) | | Unadjusted odds ratio (95%CI) | Adjusted odds ratio (95%CI) |
|---|---|---|---|---|
| | **Positive (n = 121)** | **Negative (n = 370)** | | |
| **BMI** | | | | |
| Underweight | 62 (51%) | 56 (15%) | **4.83 (3.02, 7.71)** | **2.99(1.77, 5.06)** |
| Normal | 53 (44%) | 231 (62%) | Ref | Ref |
| Above Normal | 6 (5%) | 83 (22%) | **0.32 (0.13, 0.76)** | **0.38 (0.15, 0.96)** |
| **Sex** | | | | |
| Male | 79 (65%) | 169 (48%) | **2.24 (1.46, 3.43)** | - |
| Female | 42 (35%) | 201 (54%) | Ref | |
| **Education** | | | | |
| None | 45 (37%) | 144 (39%) | 0.93 (0.61, 1.42) | - |
| PLE certificate or more | 76(63%) | 226(61%) | Ref | |
| **Employment** | | | | |
| Employed | 29 (24%) | 96 (26%) | 1.10 (0.98, 1.24) | - |
| Unemployed | 92 (76%) | 274 (74%) | Ref | |
| **Alcohol Use** | | | | |
| Never | 55 (46%) | 215(58%) | Ref | |
| < = 1drink/week | 40 (33%) | 117(32%) | 1.34 (0.84, 2.13) | - |
| > = 1 drink/week | 26 (22%) | 38 (10%) | **2.67 (1.50, 4.78)** | - |
| **Smoking** | | | | |
| Nonsmoker* | 74 (61%) | 288 (78%) | Ref | |
| Current Smoker | 47 (39%) | 82 (22%) | **2.23 (1.44, 3.46)** | 1.64(0.96, 2.78) |
| **Age(years)** | | | | |
| 15–24 | 24 (20%) | 87(24%) | Ref | |
| 25–34 | 49 (41%) | 117 (32%) | 1.52 (0.86, 2.66) | - |
| 35–44 | 29 (24%) | 94(25%) | 1.12 (0.61, 2.07) | - |
| 45–54 | 18 (15%) | 51 (14%) | 1.28(0.63, 2.58) | - |
| 55 & Above | 1 (1%) | 21 (6%) | 0.17(0.22, 1.35) | - |
| **HIV Status** | | | | |
| Negative | 75 (62%) | 241 (65%) | Ref | |
| Positive | 46 (38%) | 129 (35%) | 1.15 (0.75, 1.75) | - |
| **Symptoms** | | | | |
| Weight Loss | 94(78%) | 145 (39%) | **5.40(3.36, 8.70)** | **2.54(1.48, 4.36)** |
| Chronic Cough | 106(88%) | 213 (58%) | **5.21(2.92, 9.29)** | **3.84(2.05, 7.20)** |
| Hemoptysis | 9(7%) | 14 (4%) | 2.04(0.90, 5.19) | - |
| Fevers | 58(48%) | 95 (26%) | **2.63(1.72, 4.04)** | 1.58(0.93, 2.68) |
| Night Sweats | 56(46%) | 69 (19%) | **3.74(2.39, 5.83)** | **2.29(1.33,3.95)** |
| Chest pain | 65(54%) | 128(34%) | **2.24(1.47, 3.39)** | - |

*Former or never smoked; PLE: Primary Leaving Examinations; Ref: Reference; **In BOLD**: significant at p<0.05

patients respectively and were statistically significant independent predictors of culture-positive TB (Table 1); however, they had very low positive predictive values of 18%(95%CI: 16, 19) and 22%(95%CI: 19, 25) respectively among the overall patient population, and these PPVs were reduced further after stratifying by underweight BMI (Table 2). The least common symptom, hemoptysis, was highly specific (96%) but was present in only 7% of TB-positive patients (Table 2).

**Table 2. Accuracy of individual symptoms (comparing normal and underweight status) in predicting culture-positive TB among adult presumptive at four clinics in Kampala, Uganda.** (Prevalence = 12%).

| Symptom | n (%) | Normal weight status (n = 284) | | | |
|---|---|---|---|---|---|
| | | Sensitivity (95%CI) | Specificity (95%CI) | PPV (95%CI) | NPV (95%CI) |
| Chronic Cough | 171 (60%) | 81% (68, 91) | 45% (38, 51) | 17% (15, 20) | 94% (91, 97) |
| Fevers | 79 (28%) | 53% (39, 67) | 78% (72, 83) | 25% (19, 32) | 92% (90, 94) |
| Night Sweats | 65 (23%) | 47% (33, 61) | 83% (77, 87) | 28% (20, 36) | 92% (90, 94) |
| Weight Loss | 125 (44%) | 68% (54, 80) | 62% (55, 68) | 20% (16, 24) | 93% (90, 95) |
| Chest pain | 106 (37%) | 59% (44, 72) | 68% (62, 74) | 20% (16, 25) | 92% (90, 94) |
| Hemoptysis | 11 (4%) | 9% (3, 21) | 97% (94, 99) | 34% (14, 62) | 89% (88, 89) |
| Symptom | N (%) | Underweight status (n = 118) | | | |
| | | Sensitivity (95%CI) | Specificity (95%CI) | PPV (95%CI) | NPV (95%CI) |
| Chronic Cough | 94 (80%) | 94% (84, 98) | 36% (23, 50) | 17% (14, 20) | 98% (94, 99) |
| Fevers | 47 (40%) | 47% (34, 60) | 68% (54, 80) | 17% (11, 25) | 90% (87, 92) |
| Night Sweats | 45 (38%) | 50% (37, 63) | 75% (62, 86) | 22% (14, 32) | 92% (90, 94) |
| Weight Loss | 90 (76%) | 87% (76, 94) | 36% (23, 50) | 16% (13, 19) | 95% (90, 98) |
| Chest pain | 54 (46%) | 52% (39, 65) | 61% (47, 74) | 16% (11, 22) | 90% (87, 93) |
| Hemoptysis | 6 (5%) | 7% (2, 16) | 96% (88, 100) | 20% (5, 57) | 88% (87, 89) |

PPV: Positive Predictive Values

NPV: Negative Predictive Value

Using a score based on the number of symptoms alone provided a c-statistic of 0.77(95%CI: 0.72, 0.81) for identifying culture-positive TB. An illustrative diagnostic cutoff at three or more symptoms had an estimated sensitivity of 65% (95%CI: 56, 74) and specificity of 74% (95%CI: 69, 78), resulting in a positive predictive value (PPV) of 26% (95%CI: 22, 30) and negative predictive value (NPV) of 94% (95%CI: 92, 95) among presumptive TB patients at our study sites (Table 3).

Although underweight BMI was strongly associated with TB, underweight BMI alone was only moderately accurate for identifying TB status, with a sensitivity of 51% (95%CI: 42, 60), specificity of 84% (95%CI: 81, 88), estimated PPV of 32% (95%CI:26, 39), estimated NPV of 93% (95%CI: 91, 94), and c-statistic of 0.68 (95%CI: 0.63, 0.73).

Prediction scores that combined number of symptoms with one or more additional points for being underweight outperformed the symptom count alone. The greatest increase in discrimination (c-statistic) was achieved by assigning two additional points for underweight

**Table 3. Accuracy of number of TB symptoms (using two simple scoring systems) in predicting culture-positive TB among adult presumptive patients at four clinics in Kampala, Uganda.** (n = 491, Prevalence = 12%).

| Cutoff | No Additional Points for Underweight | | 2 Additional Points for Underweight | |
|---|---|---|---|---|
| | Sensitivity (95%CI) | Specificity (95%CI) | Sensitivity (95%CI) | Specificity (95%CI) |
| (> = 1) | 98% (93, 100) | 17% (13, 21) | 98% (93, 100) | 16% (12, 20) |
| (> = 2) | 91% (84, 95) | 47% (41, 52) | 93% (86, 97) | 41% (36, 47) |
| (> = 3) | 65% (56, 74) | 74% (69, 78) | 80% (72, 87) | 66% (61, 71) |
| (> = 4) | 43% (34, 52) | 88% (84, 91) | 69% (60, 78) | 81% (76, 85) |
| (> = 5) | 22% (15, 30) | 97% (94, 98) | 48% (39, 57) | 91% (88, 94) |
| (> = 6) | 3% (0.5, 7) | 100% (98, 100) | 24% (17, 33) | 97% (94, 98) |
| (> = 7) | N/A | N/A | 10% (5, 17) | 99% (98, 100) |
| (> = 8) | N/A | N/A | 2% (0.2, 6) | 100% (99, 100) |

**Table 4. Positive predictive value (PPV), with 95% CI, of symptom count plus two additional points for underweight status, at multiple cutoff values and multiple prevalences of TB.**

| Score Cut off | Population TB Prevalence | | | |
|---|---|---|---|---|
| | 12% (study population) | 5% | 15% | 30% |
| (> = 1) | 14% (13, 15) | 6% (5, 6) | 17% (16, 18) | 33% (32, 34) |
| (> = 2) | 18% (17, 20) | 8% (7, 8) | 22% (20, 23) | 40% (38, 43) |
| (> = 3) | 25% (22, 28) | 11% (9, 13) | 29% (26, 33) | 50% (46, 54) |
| (> = 4) | 33% (28, 39) | 16% (13, 19) | 39% (33, 44) | 61% (55, 66) |
| (> = 5) | 43% (34, 52) | 22% (16, 29) | 49% (40, 58) | 70% (61, 77) |
| (> = 6) | 51% (35, 66) | 28% (17, 43) | 57% (41, 71) | 76% (63, 86) |
| (> = 7) | 56% (30, 80) | 33% (14, 60) | 62% (35, 83) | 80% (56, 92) |
| (> = 8) | 100% | 100% | 100% | 100% |

status, resulting in a c-statistic of 0.81 (95%CI: 0.76, 0.85). In this model (as an example), a cutoff of at least four points (which could be achieved by having at least four symptoms if not underweight, or at least two symptoms if underweight) had sensitivity of 69% (95%CI: 60, 78), specificity of 81% (95%CI: 76, 85), estimated PPV of 33% (95%CI: 28, 39) and estimated NPV of 95% (95%CI: 94, 96) (Tables 3, 4, & Fig 1). The positive predictive value of this cutoff would fall to 16% (95%CI: 13, 19) in a patient population with 5% TB prevalence or increase to 61% (95%CI: 55, 66) in a population with 30% TB prevalence (Table 4).

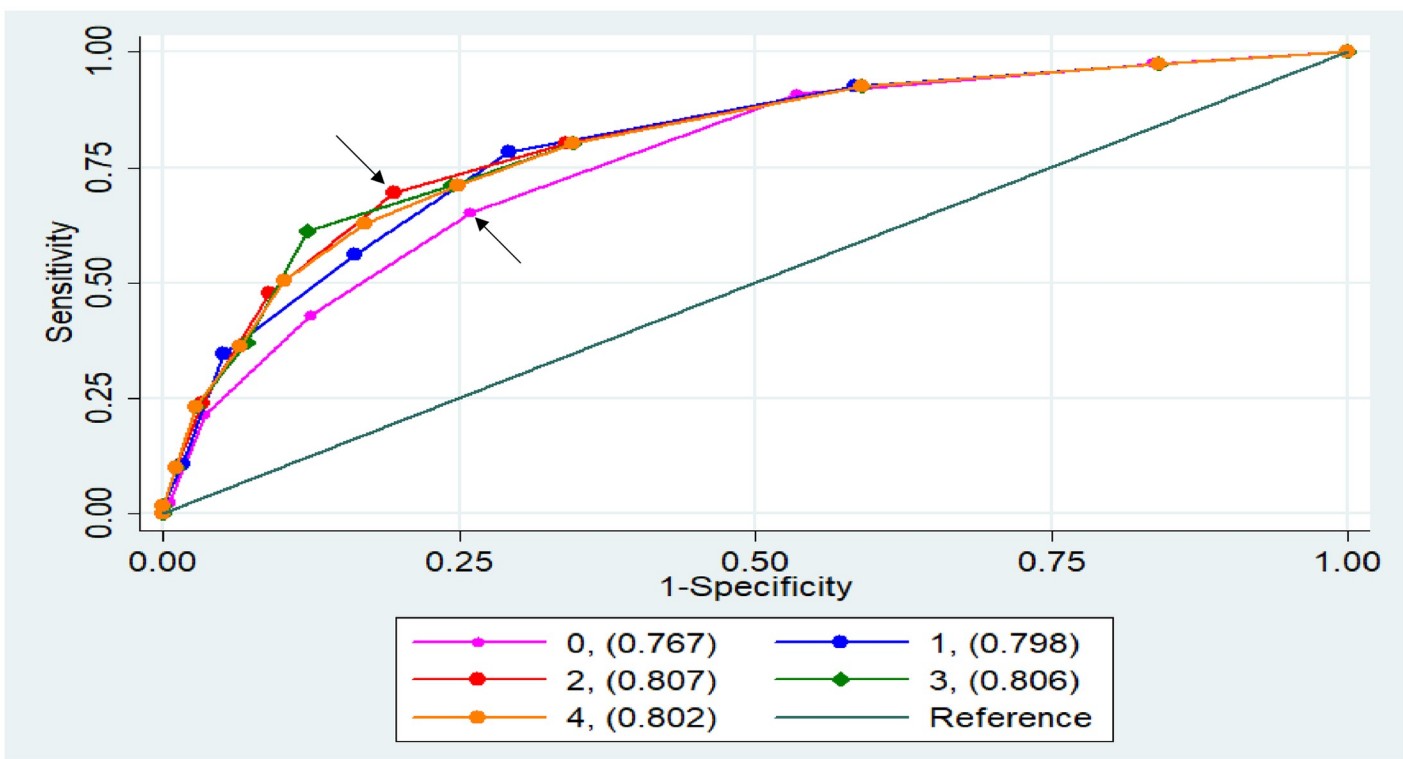

**Fig 1. ROC curves for simple scoring systems to predict culture-positive TB among patients undergoing TB evaluation.** Each score consists of the number of TB symptoms (0 to 6) reported by the patient, with additional points added if the patient had an underweight BMI. The legend shows the number of points added for underweight BMI (0 to 4), with the resulting c-statistic provided in parentheses. For example, having > = 3 symptoms had a sensitivity of 65% and a specificity of 74% for TB (lower arrow), but having a cutoff > = 4 in the combined score which added two points for underweight (a total which could be achieved by having at least four symptoms, or by having at least two TB symptoms and being underweight) had both a higher sensitivity of 69% and a higher specificity of 81% for TB (upper arrow).

## Sensitivity analysis: Symptom count and body mass index to identify Xpert-positive TB

In our sensitivity analysis that considered Xpert as the gold standard for pulmonary TB, the prevalence of pulmonary TB among the study participants was 29% (95%CI: 25, 33), and the prevalence in the underlying patient population was estimated to be 14%. The number of symptoms (without points for underweight BMI) produced a c-statistic of 0.75 (95%CI: 0.70, 0.79). Underweight BMI alone had similar accuracy for identifying Xpert-positive TB as for culture-positive TB: sensitivity 49% (95%CI: 40, 58), specificity 87% (95%CI: 83, 90), estimated PPV 38% (95%CI: 31, 46), and estimated NPV 91% (95%CI: 90, 92). A combined score with symptom number and two points for underweight status generated a c-statistic of 0.79 (95% CI: 0.74, 0.84) for predicting Xpert-positive TB (S1–S3 Tables and S2 Fig).

## Discussion

Testing delays and falsely negative test results are obstacles to ensuring that all patients with TB are treated [3, 21]. A clearer understanding of the clinical signs and symptoms most predictive of TB could help improve the early management of patients presenting for diagnosis of pulmonary TB, particularly in situations where reliable diagnostic tests are unavailable or their results are likely to be delayed. In this study, we estimated the accuracy of number of TB symptoms alone and in combination with BMI in identifying culture-positive pulmonary TB among patients presenting to health facilities in Kampala, Uganda. We found that considering both a patient's number of TB symptoms and the patient's BMI (specifically, identification of patients with underweight status) led to more accurate prediction of TB status than consideration of only the number of TB symptoms, individual BMI categories, or individual TB symptoms in isolation—even when weight loss was one of the symptoms considered. In distinguishing which patients had TB, the predictive weight of underweight status was similar to that of having two additional TB symptoms.

The accuracy of symptom-based algorithms has been best characterized for TB screening purposes (i.e., as the basis for deciding to test for TB) in general [22] and clinical [23] populations. There have been relatively few studies considering the use of symptoms or BMI to identify the patients at highest risk for TB among those who are already undergoing a diagnostic evaluation for TB. A study in South Africa [14] provided complementary results to ours, by demonstrating that BMI was useful in prioritizing patients for TB evaluation within an HIV clinic. This study, however, was restricted to patients with HIV, considered BMI as one of a suite of risk factors in a diagnostic prediction model, and considered all symptomatic patients as opposed to those whom a clinician had decided to evaluate for TB. Interestingly, within our study population, well-recognized risk factors for TB such as HIV and smoking [24–26] were not independently associated with a positive TB result among patients undergoing TB evaluation; clinicians' awareness of these associations may have already influenced their decisions about whom to test for TB, so that, less recognized indicators of risk were more useful for identifying those at highest TB risk among patients selected for testing.

The c-statistic difference that we estimated suggests that inclusion of BMI in a risk assessment could increase (by approximately 4%) the probability that a patient with true TB would be selected for empiric treatment before one without TB, beyond what could be ascertained by assessing the patients' pulmonary and constitutional symptoms alone. Such a prioritization may be needed, for example, in remote settings where diagnostic testing is often unavailable or delayed [4] or when a clinician is concerned that a patient's risk of pretreatment loss to follow up is high, presenting symptoms and conventional risk factors are often used to identify patients who may benefit from starting treatment before a bacteriologic test result is available.

Whether this estimated improvement in classification is sufficiently meaningful to merit measuring BMI in every patient likely depends on the availability of equipment (e.g., stadiometers) and the additional burden imposed on clinic staff by conducting BMI assessment. Furthermore, because underweight BMI also identifies patients at high risk for poor TB treatment outcomes and mortality [12, 27], inclusion of BMI in decisions about empiric treatment may identify TB patients who are particularly likely to benefit from prompt initiation of treatment, and therefore may add more value than the difference in c statistics suggests.

Although our study provides support for considering underweight BMI in making empiric treatment decisions, the study has some limitations. Decisions based on BMI assessment or number of symptoms have not been evaluated for their ability to improve clinical outcomes; clinical utility will depend in part on the feasibility of BMI assessment in a busy and resource-constrained clinical setting and on whether BMI assessment or symptom count adds predictive power when combined with clinician judgment. In addition, our study's reference standard of sputum culture is not perfect for determining TB status. For example, cultures may be falsely negative, some patients have extrapulmonary TB, and some patients in this study started treatment before obtaining specimens for culture, thus lowering the sensitivity of culture. It is reassuring, however, that our sensitivity analyses with an Xpert-based definition of TB provided similar results. Our quantitative results may also be biased by our study's sampling scheme, which included all patients diagnosed with TB—including those diagnosed clinically after a negative Xpert—but only a random sample of those who did not receive a TB diagnosis. Compared to other Xpert-negative patients, clinically-diagnosed patients were more likely to have suggestive TB symptoms and risk factors including low BMI, potentially leading us to underestimate the predictive power of these indicators. Finally, because we looked at urban adult patients with presumptive TB, our results may not be generalizable to rural or migratory populations or to children.

In summary, our results suggest that identification of patients with underweight BMI adds discriminatory power for identifying patients with pulmonary TB, and that routine BMI assessment should therefore be considered in settings where clinicians are often required to make decisions about empiric TB treatment for high-risk patients while awaiting bacteriologic data. Although the clinical utility of any specific algorithm would require further validation, these findings suggest that consideration of BMI could add diagnostic value above and beyond symptom count alone.

## Supporting information

**S1 Fig. Study enrollment and classification by culture status.**
(TIF)

**S2 Fig. ROC curves for a simple scoring system to predict TB, using Xpert as the reference standard.**
(TIF)

**S1 Table. Accuracy of individual symptoms (comparing normal and underweight status) in predicting Xpert-positive TB among adult presumptive at four clinics in Kampala, Uganda.**
(DOCX)

**S2 Table. Accuracy of number of TB symptoms (using two simple scoring systems) in predicting Xpert-positive TB among adult presumptive patients at four clinics in Kampala, Uganda.**
(DOCX)

**S3 Table. Positive predictive value (PPV) of symptom count plus two additional points for underweight status, using Xpert MTB/RIF as the reference standard for TB, at multiple cutoff values and multiple prevalences of TB.**
(DOCX)

**S1 Instrument. Relevant questions from participant interview.**
(PDF)

## Acknowledgments

Special thanks to the patients and staff at Alive Medical services, Kisugu Health center, Meeting point clinic and International Hospital Kampala (Touch-Namuwongo) for participating in this study.

## Author Contributions

**Conceptualization:** Peter J. Kitonsa, Emily A. Kendall.

**Data curation:** James Mukiibi, Olga Nakasolya, David Isooba, Caleb Kamoga.

**Formal analysis:** Peter J. Kitonsa, Annet Nalutaaya, Yeonsoo Baik, Katherine Robsky, David W. Dowdy, Emily A. Kendall.

**Funding acquisition:** David W. Dowdy, Achilles Katamba, Emily A. Kendall.

**Investigation:** Peter J. Kitonsa, James Mukiibi, Olga Nakasolya, David Isooba, Caleb Kamoga, David W. Dowdy, Achilles Katamba.

**Methodology:** David W. Dowdy, Achilles Katamba, Emily A. Kendall.

**Project administration:** Peter J. Kitonsa, Achilles Katamba.

**Supervision:** David W. Dowdy, Achilles Katamba, Emily A. Kendall.

**Writing – original draft:** Peter J. Kitonsa.

**Writing – review & editing:** Annet Nalutaaya, Yeonsoo Baik, Katherine Robsky, David W. Dowdy, Achilles Katamba, Emily A. Kendall.

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
