## [Decision Letter · Decision Letter 0]

9 Oct 2020

PONE-D-20-28238

Evaluation of underweight status may improve management among adults being evaluated for pulmonary tuberculosis in Kampala, Uganda.

PLOS ONE

Dear Dr. Kitonsa,

Thank you for submitting your manuscript to PLOS ONE. After careful consideration, we feel that it has merit but does not fully meet PLOS ONE’s publication criteria as it currently stands. Therefore, we invite you to submit a revised version of the manuscript that addresses the points raised during the review process.

We look forward to receiving your revised manuscript.

Kind regards,

Katalin Andrea Wilkinson, PhD

Academic Editor

PLOS ONE

Journal Requirements:

2. Please include additional information regarding the survey or questionnaire used in the study and ensure that you have provided sufficient details that others could replicate the analyses. For instance, if you developed a questionnaire as part of this study and it is not under a copyright more restrictive than CC-BY, please include a copy, in both the original language and English, as Supporting Information, or include a citation if it has been published previously.

3. In the Methods, please discuss whether and how the questionnaire was validated and/or pre-tested. If this did not occur, please provide the rationale for not doing so.

4. In your Methods section, please provide additional information about sample size determination, participant recruitment method and the demographic details of your participants. Please ensure you have provided sufficient details to replicate the analyses such as: a) approach to sample size and power calculations, b) a description of any inclusion/exclusion criteria that were applied to participant recruitment, c) a table of relevant demographic details, d) a statement as to whether your sample can be considered representative of a larger population, and e) a description of how participants were recruited.

6. Please ensure that you refer to Figure 2 and 3 in your text as, if accepted, production will need this reference to link the reader to the figure.

Reviewers' comments:

Reviewer's Responses to Questions

**Comments to the Author**

1. Is the manuscript technically sound, and do the data support the conclusions?

Reviewer #1: Yes

Reviewer #2: Partly

2. Has the statistical analysis been performed appropriately and rigorously? 

Reviewer #1: Yes

Reviewer #2: Yes

3. Have the authors made all data underlying the findings in their manuscript fully available?

Reviewer #1: Yes

Reviewer #2: Yes

4. Is the manuscript presented in an intelligible fashion and written in standard English?

Reviewer #1: Yes

Reviewer #2: Yes

5. Review Comments to the Author

Reviewer #1: Evaluation of underweight status may improve management among adults being evaluated for pulmonary tuberculosis in Kampala, Uganda.

Abstract

- Author should clearly define the main purpose of this study

- Line 32-33: Patients reported median two symptoms (interquartile range [IQR] 0, 6). The following sentence is not clear .

Materials & methods

- Could you please reorganize this section as follow?

o Study setting & design;

o study participants (population and sampling): based on fig 1 , please provide clearly explanation and you should complete the last step or line TB+ 121 & TB- 363.

o study variables

o data collection process

o Statistical analysis

o Ethical considerations (please provide the number for reference)

- Line 87-88: All individuals had their weight (in kg) and height (in meters) taken using a SECA-216 stadiometer and weighing scale (Seca Industries, Hamburg). Please revise this sentence …. It is a bit confusing.

Results

- Table 1 : Please provide the P value or mark by star the significant results and use footnotes

Conclusion:

Line 297-298: In summary, our results suggest that BMI assessment and number of symptoms should be included as part of the clinical decision-making process among adults presenting for evaluation of possible pulmonary TB.

Reviewer #2: i. Introduction:

Query 1: The author states, “Most discussions of clinical decision-making for TB center on risk factors for developing TB and on the presence and severity of characteristic symptoms. Body mass index (BMI) is often not given the same consideration but may be useful in identifying high-risk patients.”

Loss of weight is one of the main constitutional symptoms for tuberculosis (TB), and therefore this, in addition to any of the other constitutional symptoms remains a useful tool for clinical diagnosis of TB. Since weight is one of the components used to calculate the BMI, it is not surprising that low BMI will also be associated to clinical TB diagnosis. I therefore wonder if using BMI improves TB management as stated in the title or it contributes to confirming the clinical impression of TB?

As correctly stated, TB treatment centers particularly in low income countries struggle with carrying out basic clinical parameters such as weight and blood pressure, usually because of lack of availability of the necessary equipment. Thus an additional measure of height and then the calculation of BMI is very desirable but rather impractical. I am not sure about the relevance and translation of suggesting the use of BMI as an additional clinical tool for TB diagnosis.

ii. Methods:

Query 2: “We analyzed six symptoms: chronic cough (defined as cough lasting at least two weeks), fevers, weight loss (unexplained and sufficient to make clothes loose), chest pain, coughing up blood (hemoptysis), and night sweats (drenching).”

The authors measured six symptoms. I noted the omission of anorexia which is also one of the main constitutional symptoms of TB. Is there any reason why this was left out in the assessment?

What would add value is assessing a symptom or sign outside the main constitutional symptoms and evaluating this to aid in confirming clinical diagnosis of TB such as blood pressure or any other.

Query 3: “Body Mass Index (BMI) was computed as weight (kg) divided by height (meters squared) and categorized as underweight (less than 18.5 kg/m2), normal weight (18.5 to 24.9 kg/m2), overweight (25 to 29.9 kg/m2) or obese (over 30kg/m2).”

Was the BMI calculation carried out on site at the TB treatment centers? Who calculated the BMI? Knowledge of this will help in confirming feasibility. If the BMIs were calculated at the point of statistical analysis, the suggestion of using BMI in clinical care will then remain theoretical.

Query 4: “A multivariable logistic regression model was used to explore the relationship between BMI, number of TB symptoms and TB status.”

The author needs to describe these models better stating what variables were adjusted, what levels of significance were considered. Was it stepwise forward or backward?

Query 5: The study was approved by the Ethics Review Committee of the Makerere University School of Public Health, Kampala-Uganda.

Please add the ethical approval reference number. Was ethical approval sought to use data from the TB registries at the study sites?

iii. Results:

Query 6: “Participants were patients aged ≥15 years who underwent diagnostic evaluations for possible pulmonary TB at four TB Diagnostic and Treatment Units in Kampala, Uganda (including one large public clinic, two smaller private clinics, and an HIV clinic). TB evaluation at these facilities typically included sputum Xpert MTB/RIF testing [Cepheid, Inc] according to routine clinical laboratory procedures. Between 22nd May 2018 and 29th February 2020, patients who received a TB diagnosis were invited to enroll in our study; except during specified months at the largest clinic (during which all patients were eligible regardless of location of residence), enrollment was limited to residents of certain nearby zones.”

“BMI, TB symptoms, and culture status were determined for 121 patients with culture-positive TB and 363 patients with negative culture results (S1 Fig).”

The study enrolled participants over a period of almost two years in a setting where there is bound to have a high number of TB patients. The authors have not stated how many TB patients they actually screened and enrolled. They immediately report only on the number that was evaluated for sputum culture. It is also not clear how they selected the patients for sputum culture? Were there any potential biases?

Were BMIs calculated on all the patients they enrolled or for only those that had a sputum culture result?

The results presented were all based on those with a TB culture result. They didn’t describe the rest of the population. It would have been good to have a general description of the whole study population and then after focus on those with a culture result.

iv. Discussion:

Query 7: “A clearer understanding of the clinical signs and symptoms most predictive of TB could help improve the early management of patients presenting for diagnosis of pulmonary TB, particularly in situations where reliable diagnostic tests are unavailable or their results are likely to be delayed. In this study, we estimated the accuracy of number of TB symptoms alone and in combination with BMI inidentifying culture-positive pulmonary TB among patients presenting to health facilities in Kampala, Uganda.”

I am not sure that the clinical signs and symptoms most predictive for TB were assessed independently in this study. This can be done since most of the constitutional symptoms of TB were collected with the exception of anorexia. This would further confirm if weight loss or low weight independently have strong associations with a positive TB culture. If they do, then if would be sufficient to rely on them without the additional impractical measurement of a BMI. However, calculation of BMI is still highly recommended for quality clinical care.

The authors could have also explored other symptoms or signs and assessed their prediction independently or in addition to the constitutional TB symptoms. This would have added more to the existing knowledge gap.

Query 8: “Our findings support the inclusion of BMI among the clinical indicators that are considered by clinicians when they consider empiric initiation of TB treatment.”

I think this is a sweeping statement especially since we are not sure who and where the BMIs were calculated. And also because TB patients present with a history of weight loss, as well as actual low weights. Therefore, the BMI measurements of these patients are also bound to be low. I am not sure about the additional value of adding BMI measurement in already struggling health systems.

v. Conclusion:

Query 9: “Although the clinical utility of any specific algorithm would require further validation, these findings suggest that consideration of BMI adds diagnostic value above and beyond symptom count alone.”

This statement needs to be revised especially since there are methodological weakness with the choice of using BMI as main component, selection of patients assessed, and lack of clear regression model analysis description.

Query 10: It is not clear if the authors followed the STROBE guidelines when writing this manuscript? They would greatly help improve it.

6. PLOS authors have the option to publish the peer review history of their article (what does this mean?). If published, this will include your full peer review and any attached files.

Reviewer #1: **Yes: **Ghislain Poda

Reviewer #2: No

---

## [Author Response · Author response to Decision Letter 0]

20 Nov 2020

We are grateful to both reviewers, their comments and suggestions have helped us shape a more clearer manuscript in methodology, objective and results. Reviewers' comments have also informed our next steps following this research: that is for example to study the practicability of performing BMI in a busy health care setting.

---

## [Editor Report · Decision Letter 1]

24 Nov 2020

Evaluation of underweight status may improve identification of the highest-risk patients during outpatient evaluation for pulmonary tuberculosis

PONE-D-20-28238R1

Dear Dr. Kitonsa,

We’re pleased to inform you that your manuscript has been judged scientifically suitable for publication and will be formally accepted for publication once it meets all outstanding technical requirements.

Kind regards,

Katalin Andrea Wilkinson, PhD

Academic Editor

PLOS ONE
---

## [Editor Report · Acceptance letter]

1 Dec 2020

PONE-D-20-28238R1 

Evaluation of underweight status may improve identification of the highest-risk patients during outpatient evaluation for pulmonary tuberculosis. 

Dear Dr. Kitonsa:

I'm pleased to inform you that your manuscript has been deemed suitable for publication in PLOS ONE. Congratulations! Your manuscript is now with our production department. 

Kind regards, 

on behalf of

Associate Professor Katalin Andrea Wilkinson 

Academic Editor

PLOS ONE